# A Survival Analysis of Patients with Recurrent Epithelial Ovarian Cancer Based on Relapse Type: A Multi-Institutional Retrospective Study in Armenia

Lilit Harutyunyan [1,2,*], Evelina Manvelyan [3], Nune Karapetyan [1,4,5,6], Samvel Bardakhchyan [4,5,6], Aram Jilavyan [7,8], Gevorg Tamamyan [5,6,9,10], Armen Avagyan [1,2], Liana Safaryan [4,6], Davit Zohrabyan [4,6], Narine Movsisyan [1,2,11,12], Anna Avinyan [2], Arevik Galoyan [2], Mariam Sargsyan [2,5], Martin Harutyunyan [4,6], Hasmik Nersoyan [7,13], Arevik Stepanyan [7,13], Armenuhi Galstyan [7,14], Samvel Danielyan [6], Armen Muradyan [1] and Gagik Jilavyan [1,7,8]

1. Department of General Oncology, Yerevan State Medical University after M. Heratsi, 2 Koryun St., Yerevan 0025, Armenia; karapetyannune@gmail.com (N.K.); avagyanmed@gmail.com (A.A.); movses44@gmail.com (N.M.); rector@ysmu.am (A.M.); gagikjilavyan@yahoo.com (G.J.)
2. Oncology Clinic, Mikaelyan Institute of Surgery, Ezras Hasratian 9, Yerevan 0052, Armenia; dr.avinyan@gmail.com (A.A.); arevik.galoyan92@gmail.com (A.G.); mariamsargsyan364@gmail.com (M.S.)
3. Department of Reproductive Biology, University Hospitals Cleveland Medical Center, Case Western Reserve University School of Medicine, Cleveland, OH 44106, USA; evelina.manvelyan@uhhospitals.org
4. Clinic of Adults' Oncology and Chemotherapy at Yeolyan Hematology and Oncology Center, 7 Nersisyan St., Yerevan 0014, Armenia; bardakchyan-5samvel@yandex.ru (S.B.); lisafarian@gmail.com (L.S.); davzohrabyan@gmail.com (D.Z.); martinghar@gmail.com (M.H.)
5. Immune Oncology Research Institute, 7 Nersisyan St., Yerevan 0014, Armenia; gevorgtamamyan@gmail.com
6. Yeolyan Hematology and Oncology Center, 7 Nersisyan St., Yerevan 0014, Armenia; danielsamvel@yahoo.com
7. National Center of Oncology of Armenia, 76 Fanarjyan St., Yerevan 0052, Armenia; aramgagiki@yahoo.com (A.J.); hasmik.nersoyan@oncology.am (H.N.); arevik.stepanyan@oncology.am (A.S.); arminegalstyanone3@gmail.com (A.G.)
8. Department of Gynecologic Oncology, National Center of Oncology of Armenia, 76 Fanarjyan St., Yerevan 0052, Armenia
9. Pediatric Cancer and Blood Disorders Center of Armenia, 7 Nersisyan St., Yerevan 0014, Armenia
10. Pediatric Oncology and Hematology Department, Yerevan State Medical University after M. Heratsi, 2 Koryun St., Yerevan 0025, Armenia
11. Anesthesiology and Intensive Care Department, Yerevan State Medical University after M. Heratsi, 2 Koryun St., Yerevan 0025, Armenia
12. Armenian Association for the Study of Pain, 12 Kievyan Str. Apt. 20, Yerevan 0028, Armenia
13. Clinical Research and Cancer Registry Department, National Center of Oncology after V.A. Fanarjian, 76 Fanarjyan St., Yerevan 0052, Armenia
14. Diagnostic Service of the National Center of Oncology, 76 Fanarjyan St., Yerevan 0052, Armenia
*   Correspondence: lilitharutyunyan87@yahoo.com; Tel.: +37-499-403313

**Abstract:** Background: Annually, approximately 200 new ovarian cancer cases are diagnosed in Armenia, which is considered an upper-middle-income country. This study aimed to summarize the survival outcomes of patients with relapsed ovarian cancer in Armenia based on the type of recurrence, risk factors, and choice of systemic treatment. Methods: This retrospective case-control study included 228 patients with relapsed ovarian cancer from three different institutions. Results: The median age of the patients was 55. The median follow-up times from relapse and primary diagnosis were 21 and 48 months, respectively. The incidence of platinum-sensitive relapse was 81.6% (186), while platinum-resistant relapse was observed in only 18.4% (42) of patients. The median post-progression survival of the platinum-sensitive group compared to the platinum-resistant group was 54 vs. 25 months ($p < 0.001$), respectively, while the median survival after relapse was 25 vs. 13 months, respectively; three- and five-year post-progression survival rates in these groups were 31.2% vs. 23.8%, and 15.1% vs. 9.5%, respectively ($p = 0.113$). Conclusions: Overall, despite new therapeutic approaches, ovarian cancer continues to be one of the deadly malignant diseases affecting women, especially in developing countries with a lack of resources, where chemotherapy remains the primary available systemic treatment for the majority of patients. Low survival rates demonstrate the urgent need for more research focused on this group of patients with poor outcomes.

**Keywords:** recurrent ovarian cancer; platinum-sensitive relapse; platinum-resistant relapse; platinum-refractory relapse; targeted therapy; chemotherapy

## 1. Introduction

Ovarian cancer ranks third in frequency and has the highest mortality rate among gynecological cancers. In 2020, 313,959 cases of ovarian cancer were identified by Globocan (Global Cancer Observatory), and 207,252 deaths were registered [1,2]. A woman's lifetime risk of ovarian cancer is 1 in 87 [3].

According to data from the Statistical Group in the National Center of Oncology of Armenia, 197 women were diagnosed with ovarian cancer in Armenia in 2022. The morbidity and mortality rates per 100,000 of the female population were 6.6 and 2.6, respectively [4]. The first-line therapy includes debulking surgery followed by adjuvant platinum-based chemotherapy, while for patients with adverse performance status and advanced disease, treatment may start with chemotherapy and be followed by surgery [5–7]. Poor treatment outcomes are mainly explained by the high incidence of recurrence after the completion of primary treatment, which is observed in 25% of cases with early-stage diseases and in approximately 80% of cases with more advanced stages [8,9], as well as the lack of effective screening [10,11] and limited opportunities for radical surgery and systemic treatment [12].

When planning a treatment strategy for ovarian cancer relapse, accurate classification based on the interval from the end of platinum-containing chemotherapy to the first signs of the disease as either platinum-sensitive (more than 6 months) or platinum-resistant (less than 6 months) is key. [13–15]. The relapse therapy choice depends on tumor biology, the patient's ECOG status, and toxicity from previous treatment [16]. Platinum-based chemotherapy remains the backbone of the treatment; meanwhile, the use of poly (ADP-ribose) polymerase (PARP) inhibitors and anti-angiogenic (anti-VEGF) agents as maintenance therapy results in significant improvements in both progression-free survival (PFS) and overall survival (OS) [17–23].

Platinum-resistant ovarian cancer recurrences have the most unfavorable prognosis, as combined therapy with platinum-containing agents no longer leads to long-term outcomes [24–27].

There is no evidence to support the order of sequencing platinum combinations. Recommendations are limited to listing medications and their combinations, without specifying criteria for their use.

The primary endpoint of our study aimed to determine the OS and post-progression survival (PPS) of patients with relapsed ovarian cancer depending on the type of relapse, stage of the disease, and age of the patient. The secondary endpoint was to determine OS, PPS, and PFS based on the treatment regimen.

## 2. Materials and Methods

### 2.1. Study Population

The clinical material for this study were data from patients with relapsed ovarian cancer who were treated in the oncology and chemotherapy departments of the National Center of Oncology named after Fanarjyan, the Oncology Clinic of the Mikaelyan Institute of Surgery, and the Chemotherapy Clinic of Muratsan University Hospital in Armenia. Patient information was collected from medical records and outpatient cards, which were coded based on a pre-compiled coder using the scoring system displayed within it. Data regarding the current condition of patients (dead or alive) were obtained from the United Information System of Electronic Healthcare of Armenia (ArMed). A retrospective analysis of primary records of patients receiving chemotherapy for platinum-sensitive and platinum-resistant relapsed ovarian cancer was performed. This study included patients with relapsed ovarian cancer that was confirmed between 2009 and 2019, with at least a three-year follow-up after recurrence.

*2.2. Stratification According to Remission Type*

Within the framework of this study, data were examined for patients who met the following criteria: the presence of morphologically-confirmed (by the Federation of Gynecology and Obstetrics (FIGO)) stage I–IV epithelial ovarian cancer (including primary peritoneal and fallopian tube cancer), recurrence after surgery, and single or combination chemotherapy based on cisplatin or carboplatin treatment agents, either within 6 months after completing treatment (platinum-resistant) or more than 6 months after treatment (platinum-sensitive).

The remission duration and confirmation of remission status were evaluated in accordance with the accepted standards.

In line with general recommendations, we defined a complete response as the disappearance of all pathological lesions (both clinical and radiological) with the regulation of the CA-125 serum tumor marker level ($\leq$35 U/mL). A partial response was defined as partial clinical improvement with a greater than 25% (but not complete) reduction in CA-125 serum tumor marker levels and radiological regress of the tumor mass. The disease was assessed as progressive when the patient's clinical condition worsened, or the level of the tumor marker CA-125 increased by more than 25%, or radiological progression was confirmed. All other conditions were considered stable diseases. Staging was conducted according to the FIGO international staging system, which is accepted as a standard in gynecological oncology [28].

OS was measured from the time of diagnosis to death, regardless of the cause. PPS was defined as the time from the first relapse to death (any cause), and finally, PFS was calculated from the time of the first relapse to the occurrence of disease progression.

*2.3. Features and Data Analysis*

Statistical analysis of obtained data was conducted using the SPSS Statistics 23 computer program. For data analysis, depending on the type of relapse, three-year and five-year survival rates were calculated using the Kaplan–Meier nonparametric statistical method, during which the log rank (Mantel–Cox) statistical test was performed. A *p*-value < 0.05 was deemed statistically significant. Categorical variables were compared by the chi-square test.

X2 Pearson's test/Pearson's chi-squared test with Fisher's exact test was used to evaluate differences in three- and five-year survival by June 2023.

## 3. Results

*3.1. Baseline Characteristics*

3.1.1. Histologic Subtypes

Our study was conducted to analyze data from patients with epithelial malignant tumors. We did not include tumors such as granulosa cell tumors, Brenner tumors, dysgerminomas, and sarcomas, as their treatment approaches, regimens, treatment outcomes, and prognosis vary from those of epithelial tumors. Our findings showed that the vast majority of patients were diagnosed with serous adenocarcinoma (95.2%). A less common histological subtype was undifferentiated carcinoma (4% of cases), which is known to have a high recurrence rate (Table 1). Mucinous carcinoma was confirmed in 3% of patients, while endometrioid cancer was confirmed in 2%. Other histological variants, like clear cell and squamous cell cancer, were each documented in only 1% of cases.

3.1.2. Stratification According to the Primary Treatment

When analyzing the modality of the primary treatment performed in patients with relapsed ovarian cancer, the following data was obtained. The largest number of patients, 128 (56.1%), received a combination of surgery + adjuvant chemotherapy as primary treatment. For 34 (14.9%) patients, primary treatment was carried out with neoadjuvant chemotherapy + surgery, while diagnostic laparoscopy before initiation of primary treatment was performed for 66 patients, 30 (13.1%) of whom were treated with neoadjuvant

chemotherapy + surgery after laparoscopy, and 36 (15.8%) of whom were treated with surgery followed by adjuvant chemotherapy.

**Table 1.** Patients' categorization according to histological subtype.

| Histological Subtypes | Number of Patients (N = 228) | |
| --- | --- | --- |
| | Absolute | % |
| Serous adenocarcinoma | 217 | 95.2 |
| Endometrioid adenocarcinoma | 2 | 0.9 |
| Mucinous adenocarcinoma | 3 | 1.3 |
| Clear cell carcinoma | 1 | 0.4 |
| Squamous cell carcinoma | 1 | 0.4 |
| Undifferentiated | 4 | 1.8 |
| Total | 228 | 100 |

3.1.3. Stratification According to Patient Age

This study involved 228 females who had a median patient age of 55.5 years. The majority of patients (151 (66.2%)) who experienced a disease recurrence were aged 40–59 years. The number of participants over 60 y/o was 63 (27.6%). Only 14 (6.2%) patients in this study were younger, aged 20–39 y/o (Figure 1). The median follow-up time starting from the relapse was 21 (0–137) months. The median follow-up time starting from the primary diagnosis was 48 (6–347) months. According to the dataset, in the platinum-sensitive group's young age group (20–39), median OS and median PPS were 119 and 66 months, respectively; in the 40–59 age group median OS and median PPS were 54 and 24 months, respectively; and they were 51 and 19 months, respectively, in the elderly group aged >60 years old. Meanwhile, in the platinum-resistant group, the median OS and median PPS were 17 and 9 months, respectively, in the 20–39 age group, 29 and 19 months, respectively, in the 40–59 age group, and 21 and 12 months, respectively, in the age group over 60 (Table 2). A significant difference was observed between various age groups when comparing median PPSs and OSs ($p$-value = 0.01). A younger age at the time of diagnosis was associated with higher PPS. Likewise, a similar tendency was observed in terms of OS ($p$-value = 0.006) (Table 2 and Figure 2). The overall cumulative five-year survival rate for all patients in this study regardless of relapse type was 35.5%.

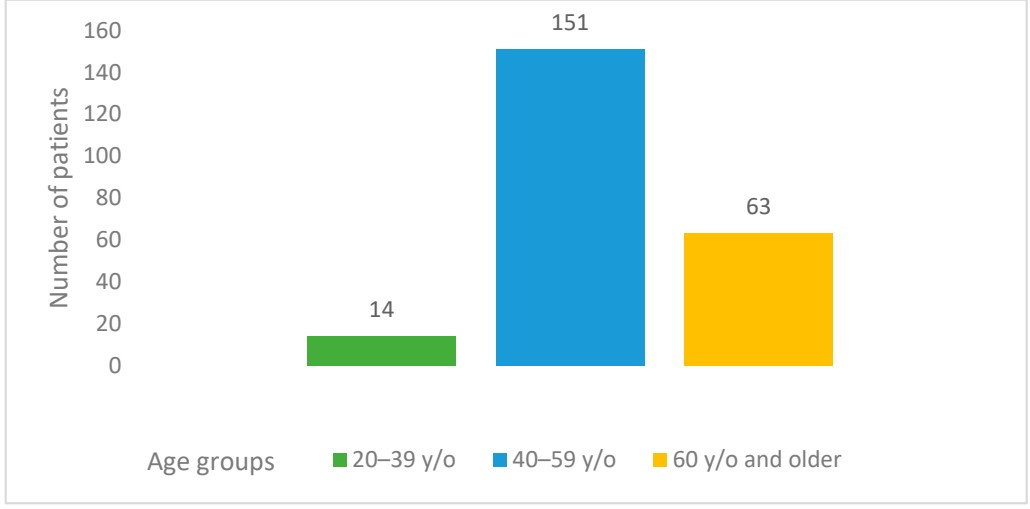

**Figure 1.** Age groups of patients with relapsed ovarian cancer.

**Table 2.** Survival rates according to patients' ages at the time of diagnosis.

| Age at the Time of dx | Survival * in the Platinum-Sensitive Group | | | Survival * in the Platinum-Resistant Group | | |
|---|---|---|---|---|---|---|
| | Median (N) # | 95% CI | *p*-Value | Median (N) | 95% CI | *p*-Value |
| 20–39 y/o | 66 (12) | 58.3; 73.7 | | 9 (2) | - | |
| 40–59 y/o | 25 (123) | 19.2; 30.8 | 0.010 | 19 (27) | 9.1; 29.0 | 0.119 |
| >60 y/o | 19 (49) | 9.2; 28;8 | | 12 (13) | 7.4; 16.6 | |
| Age at the Time of dx | Overall Survival ** in the Platinum-Sensitive Group | | | Overall Survival ** in the Platinum-Resistant Group | | |
| | Median (N) | 95% CI | *p*-Value | Median (N) | 95% CI | *p*-Value |
| 20–39 y/o | 119 (12) | 74.6; 163.4 | | 17 (2) | - | |
| 40–59 y/o | 54 (123) | 47.8; 60.2 | 0.006 | 29 (27) | 22.9; 35.1 | 0.082 |
| >60 y/o | 51 (49) | 36.4; 65.6 | | 21 (13) | 16.3; 25.7 | |

\* Post-progression survival (PPS) was calculated based on the time interval from disease relapse/progression to the last follow-up date. \*\* Overall survival (OS) was calculated based on the time interval from disease diagnosis to the last follow-up date. # Median survival was measured in months.

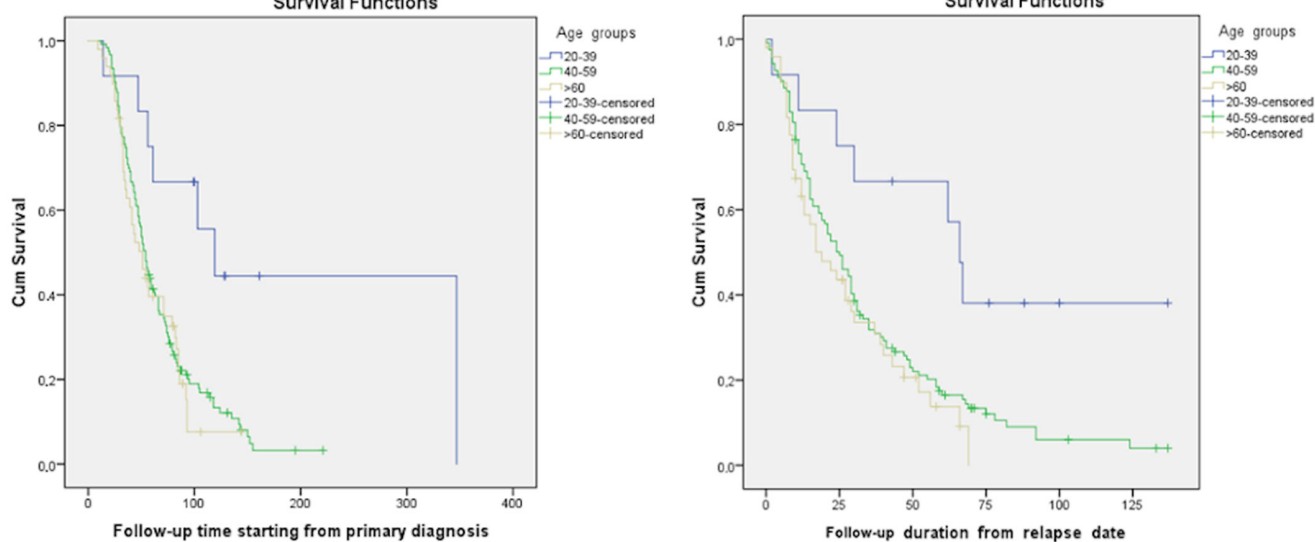

**Figure 2.** Kaplan–Maier curves based on patients' ages at the time of diagnosis in the platinum-sensitive group.

### 3.1.4. Stratification According to the Stage of the Disease

Data showed that 87.3% of patients with relapsed ovarian cancer were diagnosed with late stages (III and IV) of the disease at their primary treatment, while the proportion of early stages among patients with recurrent ovarian cancer was only 12.7%. The median OS in the platinum-sensitive relapsed group was 40 months for patients who initially had stage I disease, as shown in Table 3. In comparison, all other stage groups had a similar median OS (20–26 months) (*p*-value = 0.275).

Our findings indicated that the incidence of platinum-sensitive relapses in the study group was 81.6% (186 of 228), and only 18.4% (42 of 228) were diagnosed with platinum-resistant relapse. From provided data it became clear that the group of patients whose data was analyzed in our study was homogeneous according to key characteristics, including the stage of the disease, age, and primary treatment of relapses. This allowed us to perform statistical analysis and obtain reliable results.

**Table 3.** Survival rates according to the disease stage at the time of diagnosis.

| Disease Stage at the Time of dx | Survival * in the Platinum-Sensitive Group | | | Survival * in the Platinum-Resistant Group | | |
|---|---|---|---|---|---|---|
| | Median (N) [#] | 95% CI | *p*-Value | Median (N) | 95% CI | *p*-Value |
| Stage I | 40 (14) | 8.8; 71.2 | | 3 (2) | - | |
| Stage II | 20 (12) | 8.8; 31.3 | 0.275 | 13 (1) | - | 0.847 |
| Stage III | 24 (109) | 17.1; 30.9 | | 13 (28) | 3.9; 22.1 | |
| Stage IV | 26 (49) | 12.3; 39.7 | | 12 (11) | 7.5; 16.5 | |
| Disease Stage at the Time of dx | Overall Survival ** in the Platinum-Sensitive Group | | | Overall Survival ** in the Platinum-Resistant Group | | |
| | Median (N) | 95% CI | *p*-Value | Median (N) | 95% CI | *p*-Value |
| Stage I | 86 (14) | 63.1; 108.9 | | 13 (2) | - | |
| Stage II | 52 (12) | 47.5; 56.5 | 0.180 | 21 (1) | - | 0.499 |
| Stage III | 51 (109) | 44.0; 58.0 | | 23 (28) | 16.8; 29.2 | |
| Stage IV | 57 (49) | 47.2; 66.8 | | 34 (11) | 20.6; 47.4 | |

* Post-progression survival (PPS) was calculated based on the time interval from disease relapse/progression to the last follow-up date. ** Overall survival (OS) was calculated based on the time interval from disease diagnosis to the last follow-up date. [#] Median survival was measured in months.

Using the Kaplan–Meier method, we calculated the median survival of patients with recurrent ovarian cancer, as well as their maximum and minimum survival times. According to our data, the median OS of patients in the platinum-sensitive group compared to the platinum-resistant group was 54 months (95% CI 48.4; 59.6) vs. 25 months (95% CI 20; 30.0) (*p*-value = 0.000). Meanwhile, the median PPSs in the platinum-sensitive group compared to the platinum-resistant group were 25 months (95% CI 19.8; 30.2) and 13 months (95% CI 9.3; 16.7), respectively (*p*-value = 0.1113); three-year and five-year survival rates were 31.2% vs. 23.8% and 15.1% vs. 9.5%, respectively (Figure 3).

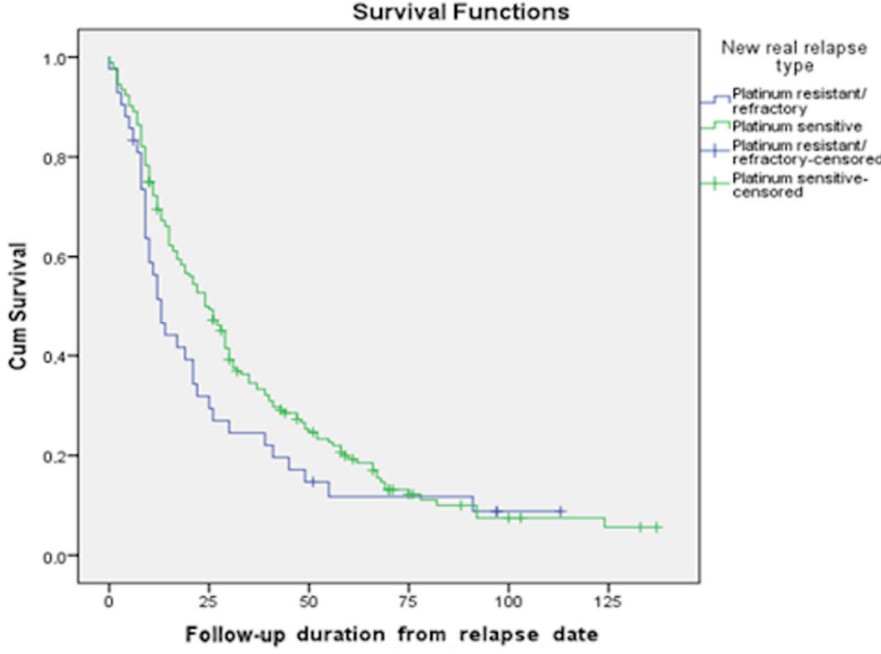

**Figure 3.** Survival functions based on the type of relapse (PPS).

*3.2. Survival after Relapse Based on the Choice of Systemic Treatment*

3.2.1. Platinum-Sensitive Relapse

We possess data regarding survival rates for the most frequently used regimens. These statistics showed that, for platinum-sensitive relapse, the most effective regimens were

found to be paclitaxel/carboplatin/bevacizumab (mPFS 28.0 months), paclitaxel/carboplatin (mPFS 26.4 months), gemcitabine/cisplatin (mPFS 19.3 months), gemcitabine/carboplatin (mPFS 17.3 months), and gemcitabine/carboplatin/bevacizumab (mPFS 14.0 months) (Figure 4).

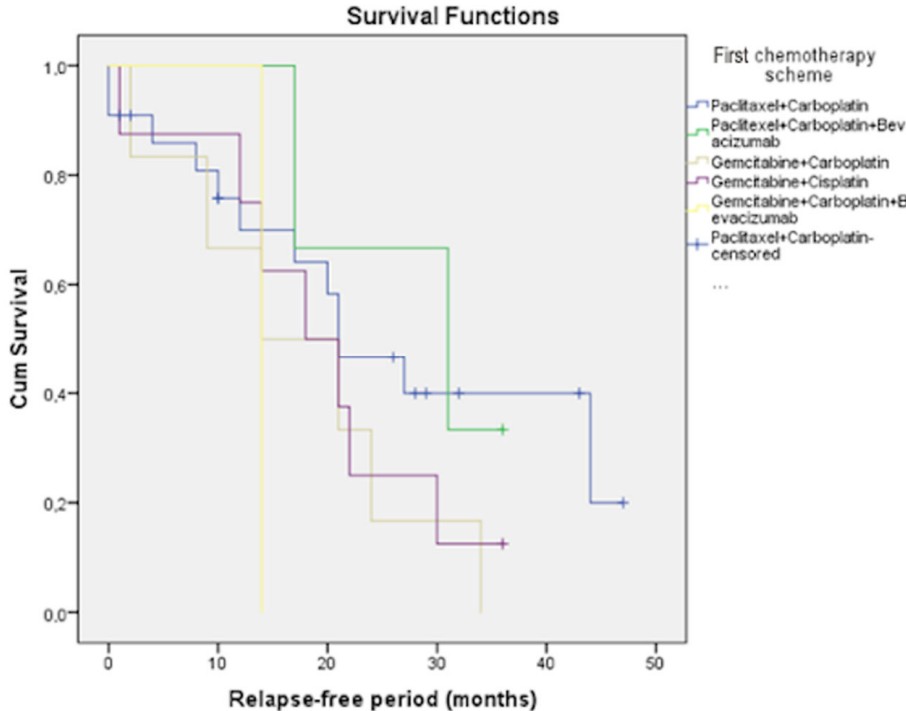

**Figure 4.** PFS in the platinum-sensitive group.

Additionally, we analyzed the results for three-year and five-year PPS for each chemotherapy regimen in both patient groups. In the platinum-sensitive group, patients' overall three-year and five-year survival rates with paclitaxel/carboplatin treatment were 29.2% and 12.5%, respectively; with paclitaxel/carboplatin/bevacizumab, 55.6% and 44.4%, respectively; gemcitabine/carboplatin, 28.6% and 0, respectively; gemcitabine/carboplatin/bevacizumab, 66.7% and 16.7%, respectively; and gemcitabine/cisplatin, 33.3% and 13.3%, respectively (Figure 5a).

3.2.2. Platinum-Resistant Relapse

We conducted a comparative analysis of the effectiveness of regimens that were administered in Armenian clinics to patients who experienced a platinum-resistant relapse with a platinum-free interval (PFI) less than 6 months. Three-year PPS rates for the analyzed groups according to the treatment they received were as follows: gemcitabine, 33%; gemcitabine/bevacizumab, 34%; gemcitabine/carboplatin, 16.7%; cyclophosphamide/doxorubicin/cisplatin, 33%; pegylated liposomal doxorubicin (PLD)/carboplatin, 50%; and paclitaxel/carboplatin, 33% (*p*-value = 0.99).

Five-year PPS rates were calculated only for patients who received paclitaxel/carboplatin (8.3%), gemcitabine/carboplatin (16.7%), and cyclophosphamide/doxorubicin/cisplatin (33%) (*p*-value = 0.79). In this group, the treatment response after relapse was reached by only seven patients; therefore, we did not include their mPFS results in this study, as the number of patients was too small (Figure 5b).

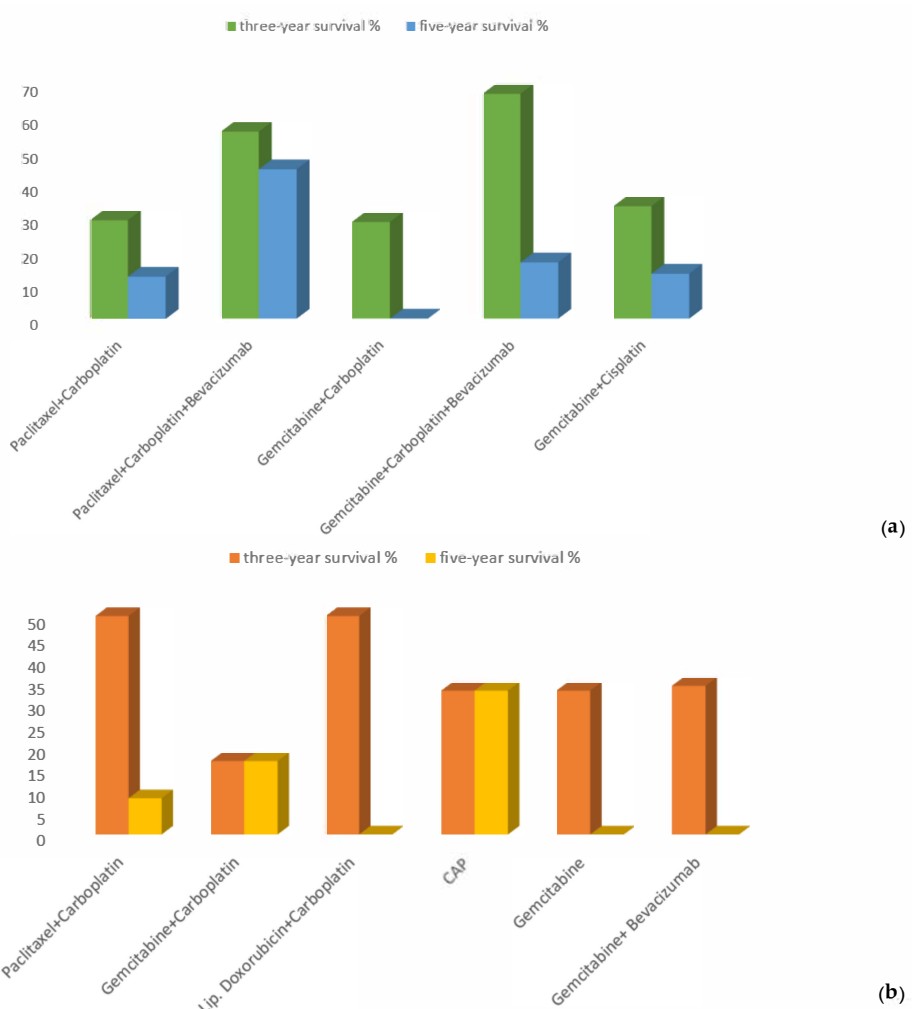

**Figure 5.** Proportions of three-year and five-year PPS in the (**a**) platinum-sensitive and (**b**) platinum-resistant groups.

## 4. Discussion

Our study showed that the majority of patients with recurrent ovarian cancer involved in the research were from 40 to 59 years old. This was probably because the median age of patients in Armenia was 55, which is considerably younger compared to statistics from the United States, where the median age for this disease is 63 [29], and also lower than that reported in Japanese studies, where the median age of patients at relapse was reported as 59 [30]. Conversely, in India, ovarian cancer is present in a younger age group, with a median age < 55 years reported by most studies, and even <50 years in some regions of the country [31]. This phenomenon might be attributed to underdiagnosis and undertreatment of ovarian cancer among the elderly population in limited-source settings. Our data demonstrated that the best OS and PPS were detected in the younger age group of the platinum-sensitive group, which was twice as high (119 months) compared to those of other age groups. These results correlate with statistics in the literature regarding age-connected mortality in ovarian cancer: the older the patient is, the worse the survival rate [32]. However, when we assessed the platinum-resistant group, the picture was different—the middle-aged group (40–59) exhibited longer survival, while the elderly and young adult groups had relatively the same survival rates. These non-intuitive results might be attributed to the small sample size in the platinum-resistant group.

As the results revealed, the majority of patients with ovarian cancer had serous adenocarcinoma (95.5%), which is a much higher rate than in Eastern European countries like Poland, where the incidence of high-grade serous adenocarcinoma is approximately

74% [33]. One of the possible explanations for this difference could be the fact that our study included only patients with recurrence, which may be more frequent in the serous adenocarcinoma subtype.

According to our study results, the vast majority of patients with relapsed ovarian cancer initially had advanced FIGO stages III and IV, and the proportion of early stages among patients with recurrent ovarian cancer was approximately 13%. These data once again confirm the fact that disease stage is the most unfavorable prognostic factor in ovarian cancer, emphasizing the importance of early detection [34]. Interestingly, the median survival after relapse in the platinum-sensitive relapsed group was twice as good (40 months) for patients who initially had stage I disease. In comparison, all other stage groups had similar survival rates (20–26 months). These data differ from the results of the article published by Rajendra Kumar Meena in 2022 in JCO Global Oncology, according to which stage I and II patients have better survival than more advanced-stage groups do [31]. An Italian study that was carried out by Gadducci et al. also showed statistical significance between survival after recurrence and initial clinical stage (I, IIA vs. IIB–IV) [35].

According to our study results, regardless of the type of relapse, the cumulative five-year OS for Armenian patients was 35.5%, which was lower compared to that of Sudan (38%), which is classified as a low-middle-income country [36]. Our data showed that more than 80% of patients with ovarian cancer relapse had platinum-sensitive disease, so the majority of them had a chance to be treated with a platinum combination again, in the absence of comorbidities [5,8,13,14]. As a result, three- and five-year PPS rates of platinum-sensitive relapse were almost twice as high as those of the platinum-resistant type were. These results are similar to those of the study conducted by Ai Miyoshi et al., which showed that the patients who had a relapse with a treatment-free interval (TFI) of less than 6 months had notably worse outcomes than those with a TFI exceeding 6 months [30].

Platinum-sensitive relapse is considered a chemo-sensitive disease in more than half of patients [37]. As demonstrated in our study results, paclitaxel/carboplatin was one of the most effective regimens for platinum-sensitive relapsed ovarian cancer. This was also the main regimen patients received during their primary treatment. One of the biggest advantages of this regimen is its affordability and availability. Our results are in line with the generally accepted guidelines, showing that rechallenging the "backbone" regimen remains one of the most effective treatments in cases of recurrence [13–18]. A generally accepted standard of care is readministration of a platinum agent combined with one of the following medications: PLD, paclitaxel, or gemcitabine, with or without a VEGF inhibitor like bevacizumab [38,39].

Based on our analysis, adding bevacizumab to the chemotherapy regimen resulted in higher three-year and five-year PPS, which was also shown in the AURELIA trial published in the Journal of Clinical Oncology (JCO) by Pujade-Lauraine et al. [40]. The shortest survival rates were observed in patient groups that did not receive platinum compound therapy without having a chance to reach three- and five-year OS.

One of the goals of our study was to evaluate the effectiveness of a range of regimens in the treatment of platinum-resistant relapses and their impact on three- and five-year PPS rates. The generally accepted standard of care for platinum-resistant ovarian cancer is single-agent chemotherapy with non-platinum agents like gemcitabine, docetaxel, paclitaxel, topotecan, PLD, etc. [41–43].

As we have already demonstrated, the three-year PPS for patients with platinum-resistant relapses was 23.8%, while their five-year PPS was only 9.5%. The primary monotherapy regimen that exhibited better survival rates was gemcitabine. Our study did not show a survival benefit with bevacizumab in this group of patients. Interestingly, some patients received platinum compounds during their relapse treatment even though their platinum-free intervals were less than 6 months, which is not accepted as a standardized approach. Nevertheless, only these particular groups of patients (treated using paclitaxel/carboplatin, gemcitabine/carboplatin, or cyclophosphamide/doxorubicin/cisplatin) reached five-year survival after relapse. This can be explained by the retrospective analysis conducted by the

Australian Ovarian Cancer Study. According to the latter, PPS improved after platinum-based chemotherapy, even for patients with a platinum-free interval of less than 6 months (the median PPS was 17.7 months after platinum-based chemotherapy vs. 10.6 months after a non-platinum regimen). However, a platinum-free interval of more than 6 months does not necessarily guarantee a response to future platinum-based chemotherapy [18,44].

It should be recognized and emphasized that, especially in a developing country like Armenia, the availability of and accessibility to medications often have a significant impact on the choice of anti-relapse treatment. There is a lack of treatment accessibility due to insufficient government coverage and limited availability of essential medications [45].

This study had certain limitations, including the following.

- Due to its retrospective design, there was a notably inadequate documentation of important details in medical records, particularly regarding treatment-related toxicity.
- Most patients who relapsed received only chemotherapy due to a lack of access to the targeted therapy. A number of targeted agents, like bevacizumab, PARP inhibitors (olaparib, rucaparib, and niraparib), immunotherapy (pembrolizumab and dostarlimab), and folate receptor alfa inhibitor marvetuximab soravtansine-gynx, are not affordable for Armenian patients. Except for bevacizumab, which was available for a few patients, all other abovementioned agents are not even registered in Armenia.
- There was a small cohort of patients in the platinum-resistant group.
- In our study, we did not take into consideration the impact of "second look" surgeries during the treatment of recurrences.

## 5. Conclusions

The present analysis will contribute to the improvement of treatment outcomes for patients with relapsed ovarian cancer. Overall, despite new therapeutic approaches, ovarian cancer continues to be one of the deadly malignant diseases affecting women, especially in developing countries with a lack of resources, where chemotherapy is still the primary available systemic treatment for the majority of these patients. Persistently low survival rates highlight the urgent need for more research regarding this group of patients with poor outcomes.

**Author Contributions:** L.H.—project administration, data curation, supervision, visualization, writing—original draft, writing—review and editing, and investigation. E.M.—visualization, conceptualization, and writing—review and editing. S.B.—visualization, conceptualization, and writing—review and editing. N.K.—visualization, conceptualization, and writing—review and editing. A.J.—data curation and writing—review and editing. G.T.—visualization and writing—review and editing. A.A. (Armen Avagyan)—writing—review and editing. L.S.—writing—review and editing. D.Z.—writing—review and editing. N.M.—writing—review and editing. A.A. (Anna Avinyan)—writing—review and editing. A.G. (Arevik Galoyan)—writing—review and editing. M.S.—writing—review and editing. M.H.—writing—review and editing. H.N.—data curation and investigation. A.S.—data curation and investigation. A.G. (Armenuhi Galstyan)—writing—review and editing. S.D.—conceptualization and writing—review and editing. A.M.—conceptualization, project administration, supervision, and visualization. G.J.—conceptualization, project administration, supervision, visualization, and writing—review and editing. All authors have read and agreed to the published version of the manuscript.

**Funding:** This research received no external funding.

**Institutional Review Board Statement:** This study was approved by the Ethics Committee of Yerevan State Medical University (Code N 5-4/21, date 16 December 2021).

**Informed Consent Statement:** Informed consent was waived, as this study was a retrospective audit of medical records.

**Data Availability Statement:** Data that support the findings will be available in the repository at URL [https://figshare.com/s/b9c0f3ab18b666fa523c] (accessed on 9 February 2024).

**Conflicts of Interest:** L.H. reports lecture fees and travel support from Roche and Novartis Pharmaceuticals outside of the submitted work.

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
