# Peer review of "A Survival Analysis of Patients with Recurrent Epithelial Ovarian Cancer Based on Relapse Type: A Multi-Institutional Retrospective Study in Armenia"

_curroncol, doi:10.3390/curroncol31030100_

Round 1

Reviewer 1 Report

Comments and Suggestions for Authors

In the paper, the authors detected the difference of prognosis between chemotherapy resistant and sensitive recurrent ovarian cancers. They could collect a certain number of cases with recurrent ovarian cancers only in one hospital. They could show the concrete survival rate of patients with recurrent ovarian cancers and focus on the severe cases.

I think this report is meaningful, because ovarian cancers are severe disease. So, I have one request.

I want you to classify the cases by the types, including serous carcinoma, mucinous carcinoma, clear cell carcinoma, endometrioid carcinoma and others, since the prognosis is different. At least, you should summarize the number of each case in a new table.

References are appropriate.

Author Response

Dear Reviewer,

Thank you for your comments.

We have made all the appropriate changes and added the required data to the manuscript.

Please kindly find the attachment.

Best regards,

Lilit Harutyunyan

Reviewer 2 Report

Comments and Suggestions for Authors

The authors performed a standard study to report the survival outcome of relapsed ovarian cancer patients in Armenia.

Major comments:

1) The way to study the survival after relapse based on the choice of the systemic treatment is nice and meaningful. Please present the data (3.2.1 and 3.2.2) by graphic.

2) In the discussion, the authors compared the Armenia's data with United States's and India's data. How about to compare the data from Mid-east and Eastern European countries?

Author Response

(The authors gave the same response as above.)

Reviewer 3 Report

Comments and Suggestions for Authors

The article is interesting and could bring new insight for developing countries, which do not have access to targeted therapy. However the article has many flaws:

1. The introduction is too long! According to guidlines for writing it should be no more than 300 words. 

2. English need corrections by a native speaker . There are many strange phrases - " The risk of getting ovarian cancer for women during their ".... "the chance of dying from ovarian cancer "....

3. I do not see detail information about primary surgery! In what percentage of patients, who underwent debulking surgery- RO (no mascoscopic residual disease) was achieved? You should  compared patients only with RO or R1-R1, as results you mentioned for relapses are expected if you compared R0 to R1 

4.  Too complicated sentence. Please revised "According to the data, from the platinum-sensitive arm, in the age group from 20 to 39, median overall survival and median survival after relapse were 119 months and 66 months, in the age group from 40 to 59: 54 months and 24 months, and finally in the elderly group with the age more than 60 years old, 51 months and 19 months, respectively"

5. Surgery is rarely mentioned in the manuscript - What proportions of patients with relapes had RO, R1? What proportion of patients underwent neoadjuvant chemo followed by surgery? Optimal debulking together with chemo is the most important prognostic factor and data for the surgery is lacking

6. Relapses aslo depends on histological type ? What is the histological type of the tumors?

7. Figure 1 - such figures are no longer used. It should be changed

8. Is figure 3 needed? It could be stated in one simple sentence! 

9. In limitations - it should be mentioned the small cohort of overall patients in the study

10. Please, correctly use abbreviations! 

Comments on the Quality of English Language

English native speaker is needed!

Author Response

(The authors gave the same response as above.)

Round 2

Reviewer 1 Report

Comments and Suggestions for Authors

Thank you for responding to my requests (table 1). I have no other request.

Reviewer 2 Report

Comments and Suggestions for Authors

The authors have responded my comments well.

Reviewer 3 Report

Comments and Suggestions for Authors

Thank you for your revisions

Comments on the Quality of English Language

English grammar is fine! Some minor remarks